# Analysis of Physiological Parameters and Driver Posture for Prevention of Road Accidents: A Review

**DOI:** 10.3390/s25196238

**Published:** 2025-10-08

**Authors:** Alparslan Babur, Ali Moukadem, Alain Dieterlen, Katrin Skerl

**Affiliations:** 1Institute of Technical Medicine, Faculty of Health, Medical and Life Sciences, Furtwangen University of Applied Sciences, 78120 Furtwangen, Germany; alparslan.babur@hs-furtwangen.de (A.B.); katrin.skerl@hs-furtwangen.de (K.S.); 2Institute for Research in Computer Science, Mathematics, Automation and Signalling, University of Haute Alsace, 68100 Mulhouse, France; ali.moukadem@uha.fr

**Keywords:** physiological parameter measurement, posture recognition, pressure sensing mat, heart rate, respiration rate, machine learning

## Abstract

This review provides an overview of existing accident prevention methods by monitoring the persons’ physiological state, observing movements, and physiological parameters. Firstly, different physiological parameters monitoring systems are introduced. Secondly, various systems dealing with position recognition on pressure sensing mats are presented. We conduct an in-depth literature search and quantitative analysis of papers published in this area and focus independently of the application (drivers, office and wheelchair users, etc.). Quantitative information about the number of subjects, investigated scenarios, sensor types, machine learning usage, and laboratory vs. real-world works is extracted. In posture recognition, most works recognize at least forward, backward, left and right movements on a seat. The remaining works use the pressure sensing mat for bedridden people. In physiological parameters measurement, most works detect the heart rate and often also add respiration rate recognition. Machine learning algorithms are used in most cases and are taking on an ever-greater importance for classification and regression problems. Although all solutions use different techniques, returning satisfactory results, none of them try to detect small movements, which can pose challenges in determining the optimal sensor topology and sampling frequency required to detect fine movements. For physiological measurements, there are lots of challenges to overcome in noisy environments, notably the detection of heart rate, blood pressure, and respiratory rate at very low signal-to-noise levels.

## 1. Introduction

Sitting is a common activity in our daily lives. Motor vehicle drivers in particular spend almost their entire working time seated behind the wheel. According to the German Federal Statistical Office, the number of car accidents in Germany increased by more than 11 % between 2010 and 2019 [1]. Drowsiness and fatigue are among the main factors that threaten road safety, leading to severe injuries, fatalities, and economic losses. Increased drowsiness results in reduced driving performance. The lack of alertness caused by the unconscious transition from wakefulness to sleep is responsible for numerous serious road accidents [2]. Since 2010, the annual number of car accidents in Germany has risen from 2.41 million to 2.68 million [1]. However, a decrease was observed in 2020 due to the pandemic-related lockdown (see Figure 1).

Road traffic accidents are generally classified into two main categories: property damage and personal injury. In Germany, nearly 2000 road traffic accidents involving personal injury occur each year, with an increasing trend that can largely be attributed to driver fatigue [1]. Driver fatigue may result from physical exertion (active fatigue), monotonous driving tasks (passive fatigue), or sleep-related causes [3]. In the literature, five principal approaches to the detection of driver fatigue are described [3]:**Subjective reporting:** The driver completes a self-assessment questionnaire that can be used to determine the fatigue levels.**Biological Feature-Based Driver Fatigue Detection:** Biological signs offer good indication of early onset of fatigue and can be utilized to prevent accidents. Electroencephalography (EEG), electrocardiogram (ECG), and electrooculography (EOG) are commonly seen examples.**Physical Feature-Based Driver Fatigue Detection:** Physical feature-based systems for fatigue detection can be divided into techniques based on eyes, mouth, and face/head. Mostly the installation of a camera is mandatory.**Vehicular Feature-Based Fatigue Detection:** Vehicular features like lane crossing, steering wheel angle pressure changes on brake and accelerator, vehicle speed, load distribution on the driver’s seat, etc., are mostly used in practice.**Hybrid Feature-Based Fatigue Detection:** The methods mentioned above are combined.

Accidents caused by driver fatigue exhibit a higher fatality rate and result in more severe damage to the surrounding environment compared to accidents involving alert drivers [3,4]. Figure 2 illustrates that the economic costs of road traffic accidents in Germany totalled EUR 33.90 billion in 2019, of which EUR 12.86 billion were attributable to personal injuries. For comparison, the costs of personal injuries in 2010 amounted to EUR 12.36 billion [1]. Continuous monitoring of the driver’s physiological and cognitive state has the potential to mitigate the number of road traffic accidents.

In general, three main classes of parameters influence accident risk: the condition of the vehicle, the driving environment (e.g., traffic density, weather conditions), and the physiological state of the driver. The condition of the vehicle is primarily determined by the manufacturer and influenced by the availability of driver assistance systems. The driving environment, by contrast, is shaped by external factors beyond direct control. The physiological state of the driver, however, can be monitored and potentially improved through targeted interventions offered by third-party providers. While the discussion of accident statistics serves as contextual motivation, the focus of this review lies in the detection of physiological parameters of the driver. Modern vehicle seats have become increasingly sophisticated, as manufacturers innovate to enhance not only comfort but also occupant well-being and safety [5].

Comfort: heating, ventilation, massage, lumbar support;Safety: 3-axis adjustment, presence detection.

Studies have demonstrated that accidents can be prevented through the continuous monitoring of a person’s health condition [6,7,8,9]. Several systems for health condition monitoring are already available on the market [10,11,12,13,14,15]; however, most of these rely on camera-based approaches. Such solutions may cause discomfort, as individuals may feel observed, and they also raise concerns regarding data protection and potential misuse of personal information due to insufficient security measures. Alternative systems focus on unobtrusive monitoring methods, such as posture and pressure recognition, to detect sleeping or sitting positions [16,17,18,19,20,21,22], which are closely interrelated [5]. The use of a PSM to record and analyze pressure distribution represents an effective approach to address these challenges.

Several review papers have been published recently, such as [23,24]. In [23], the authors focused on emerging technologies for RGB-D cameras and deep learning, whereas [24] aimed to provide an overview of recently implemented driver drowsiness detection systems. However, the aspect of physical feature-based driver fatigue detection received comparatively little attention. For example, Feng et al. [25] proposed an authority allocation strategy for shared steering control that dynamically adjusts the distribution of steering authority between driver and automation based on mutual trust levels. Their hierarchical framework combines fuzzy rules and model predictive control to optimize both smooth control transitions and lane-keeping performance, validated through driver-in-the-loop experiments. This line of research highlights the relevance of considering driver state and trust in the development of preventive systems for reducing accidents related to fatigue or atypical health conditions. Fang et al. [26] sought to reduce accidents by developing a human–machine shared control framework that considered time-varying driver characteristics, such as vehicle and steering wheel velocity and acceleration. While their work concentrated primarily on the vehicle, the driver was considered only to a limited extent.

This is precisely where the present review distinguishes itself. Despite a growing body of research on driver fatigue detection, relatively few studies have investigated unobtrusive, pressure-based approaches for monitoring physiological parameters and posture in seated drivers. Existing reviews predominantly address camera-based systems or general drowsiness detection, leaving a gap in the understanding of how PSMs can be utilized for real-time, privacy-preserving driver monitoring. The objective of this review is therefore to synthesize the current state of research on PSM-based methods, identify prevailing trends and limitations, and provide recommendations for future work in this underexplored domain.

In the current stage of our research, we employed machine learning (ML) algorithms, including classification methods for static sitting postures and regression models for dynamic driver movements recorded on a PSM. To address the high dimensionality of the temporal data, signal processing techniques such as feature extraction and dimensionality reduction were applied. Furthermore, we utilized a diverse participant cohort and systematically evaluated the impact of previously known versus novel users on model generalization performance.

Our research revealed a noticeable lack of studies addressing driver fatigue detection using PSMs. Therefore, this work first provides a state-of-the-art review of existing approaches for monitoring physiological parameters and recognising movements in seated individuals. Building on this foundation, we present an overview of posture recognition methods specifically utilising PSMs. To the best of our knowledge, this review is the first to systematically integrate posture recognition and physiological monitoring using PSMs for driver fatigue detection, providing a structured classification of methods, highlighting methodological gaps that limit comparability, and outlining research opportunities to support the development of preventive, real-time, privacy-preserving driver monitoring systems.

## 2. Materials and Methods

The literature was primarily identified using Google Scholar with the following keywords: “pressure sensor mat”, “posture recognition”, “vital signs measurement”, “sleepiness detection”, “fatigue detection”, “Electrocardiography”, “pressure distribution on a seat”, “driver fatigue monitoring system”, “driver fatigue pressure mat”, and “pattern recognition using ML”. No Boolean operators or database-specific search strings were applied. Additional sources were identified from the references cited in relevant articles.

Initially, articles published in English from 2016 onwards were considered to capture recent developments. Subsequently, additional studies published prior to 2016 that were deemed relevant were also included.

Articles were included if they reported experimental or observational studies (qualitative or quantitative, prospective or retrospective) addressing either of the following:Measurement of physiological parameters such as postural sway, fatigue, sleepiness, and vital signs, orPosture recognition using PSMs in combination with ML algorithms.

In the category of physiological parameter measurement, participants were predominantly drivers; however, some studies included other participant groups depending on the focus of the respective research. For posture recognition using PSMs, studies typically involved small and diverse participant cohorts, reflecting the exploratory nature of research in this field.

Articles were initially screened by title and abstract, with full-text review conducted if deemed relevant. In total, 12 studies were included for physiological parameter measurement and 9 studies for posture recognition using PSMs. Studies that primarily focused on eye-state monitoring or relied on camera-based methods were excluded to ensure data security and prevent potential misuse of personal information.

Limitations: This review does not follow a formal systematic review protocol. Only Google Scholar was searched, keywords were applied individually without Boolean operators, and study selection was based on perceived relevance rather than pre-specified criteria. Consequently, some relevant studies may not have been identified.

This paper is structured as illustrated in Figure 3. We begin by presenting methods for measuring physiological parameters, distinguishing between obtrusive and unobtrusive approaches. The findings are summarized in Table 1, which details the sensor type, sensor placement, monitored parameters, study objectives, and detection methods, organized by sensor category: electronic sensors first, followed by vision-based sensors, and finally pressure sensors. Subsequently, the focus shifts to posture recognition using PSMs. The corresponding results are presented in Table 2, providing a detailed analysis of sensor placement, number of sensors, recognised postures, classification techniques, and achieved accuracies.

## 3. Physiological Parameters Measurement

Today, the requirements for technical innovations are increasingly diverse. The measurement of physiological signals is widely employed for detecting potentially life-threatening situations [42,43,44]. The physiological signals considered in this review include postural sway, fatigue, sleepiness, and vital signs. Vital signs are regarded as clinically significant, as they indicate life and provide reliable information about a patient’s current and future health status. The key vital signs are blood pressure (BP), body temperature (BT), heart rate (HR), and respiration rate (RR) [42]. Typically, vital signs such as HR and RR are measured using wired connections; however, in certain contexts, these connections may interfere with the driver [21]. Consequently, some measurements must be conducted unobtrusively. The reviewed papers were classified according to whether the measurement technique was obtrusive or unobtrusive, as summarized in Table 1.

### 3.1. Unobtrusive Physiological Parameter Measurement

In this section, the reviewed papers employed unobtrusive techniques to extract physiological parameters. In our research, we also identified additional studies on physiological parameter measurement, which were not included in Table 1 [9,15,21,45], either because they were outdated or no longer considered state-of-the-art. Lyra et al. investigated both skin temperature trends and respiration-related chest movements to determine RR using low-cost hardware in combination with advanced algorithms [12]. For this purpose, a deep learning-based algorithm was implemented for real-time vital sign extraction from thermography images. The primary parameters monitored were HR, BP, RR, and BT, providing information about the general physical status [42]. A clinical trial was conducted in an intensive care unit, recording data from a total of 26 patients using the infrared camera Optris PI 450i at four frames per second. An optical flow algorithm was employed to extract RR from the chest region. The results indicated a mean absolute error of 2.69 bpm. While the method allows real-time vital sign extraction on a low-cost system-on-module, it still requires the installation of a camera for monitoring.

Nakane et al. focused on fatigue detection by routinely monitoring an individual’s health to prevent accidents and illnesses [17]. Their study concentrated on posturography in the seated position, as sitting is a common daily activity and can be measured without restricting the user’s movement. The objective of the experiment was to investigate how postural sway in seated subjects differs between fatigue and non-fatigue conditions. The PSM used in the study comprised 256 pressure sensors arranged in a 16 × 16 reticular pattern, with a sampling rate of 10 Hz. Experimental data were collected from four participants, with measurements taken over seven to eight days per participant. It was assumed that subjects would exhibit greater fatigue in the evening than in the morning. To assess fatigue, the centre of pressure was calculated once in the morning and once in the evening. A positive difference in the centre of pressure between morning and evening indicated higher fatigue in the evening, while a negative difference suggested lower fatigue. The study demonstrates that fatigue may potentially be detected using a PSM; however, due to the small sample size, further experiments are required for a more robust assessment.

Tu et al. [33] presented a method for detecting fatigue by analyzing the pressure distribution of individuals on a seat. Features were computed across multiple domains—time, frequency, and time–frequency—from the pressure signals, and feature selection was applied to identify the most informative subset. For evaluation, seven ML models (kNN, SVM, Decision Tree, Random Forest, Adaboost, XGBoost, LightGBM) were trained on both the full feature set and the selected subset. These models achieved fatigue detection accuracies above 92%, with Random Forest demonstrating the highest performance. However, several limitations were noted. Data diversity was limited, as interpolation for augmentation could introduce noise. The small dataset also resulted in class imbalance, reducing model robustness and necessitating techniques such as oversampling or cost-sensitive learning. Furthermore, fatigue labels were based on self-reported scores using the Karolinska Sleepiness Scale (KSS), which may introduce bias.

Lima et al. developed a contactless, low-cost vital sign monitoring system for cardiac measurements in elderly individuals [27]. A low-cost strain-gauge sensor was positioned at the top end of the mat, aligned along the foot–head axis, which maximized the cardiac component while reducing the respiration magnitude in BCG measurements. This configuration allowed higher HR recognition by the ML algorithm. ECG (Electrocardiography) measurements, obtained using an AD8232 Heart Monitor, were synchronized with BCG recordings via an Arduino Uno. For seated individuals, strain-gauges were replaced by piezo disks, which respond only to changes in movement and do not produce an offset signal due to the person’s weight. Different body positions generated distinct signals, enabling the determination of a person’s posture based on characteristic patterns, as illustrated in Figure 4. After BCG measurement, respiratory and cardiac signals were extracted using a Butterworth band-pass filter with cut-off frequencies of 0.05 and 0.5 Hz for RR. HR was extracted using low-pass and high-pass finite impulse response filters with cut-off frequencies of 20 Hz and 0.5 Hz and orders of 80 and 540, respectively. HR was determined using the angular dissimilarity function, while RR was obtained via a peak detection function [27]. The results demonstrate that accuracies of up to 98% can be achieved in the supine position using strain-gauge sensors, with 88.6% in the prone position, 96.6% when lying on the left side, and 98% when lying on the right side.

Huang et al. explored the use of a PSM for RR and HR estimation [34]. Two methods incorporating body morphology information, based on shoulder blades and weighted centroid, were developed for RR calculation. In both RR and HR estimation, a growth region algorithm was applied to separate the torso from the arms. HR was estimated by dividing the segmented image into 3 × 3 square blocks, with the nine sensor values summed into a single vector for the entire three-minute recording period. Only values exceeding a predefined threshold were considered. Each signal was subsequently filtered using a fourth-order Butterworth filter, and the short-time Fourier transform was applied to the signals. Experimental data were collected from 15 participants in the supine position using a PSM placed beneath the upper torso. RR and HR estimations derived from accelerometer sensors attached to participants’ bodies were used as reference measurements to evaluate the accuracy of the proposed methods. The sensor used for data acquisition was the CONFORMat system (Tekscan Inc., Boston, MA, USA), a bi-dimensional pressure mapping sensor containing a 42 × 48 array of discrete pressure sensors with 8-bit signal quantization and a sampling rate of 50 Hz.

### 3.2. Physiological Parameter Monitoring During Driving Environments

In this section, the reviewed papers applied techniques primarily based on ECG to analyze physiological parameters. In their study, Leicht et al. evaluated the informative value of capacitively coupled ECG (cECG) for patients with pre-existing heart conditions during a naturalistic driving simulation [35]. The addressed research questions included a comparison of cECG and reference ECG (rECG) regarding signal quality, artefact rate, proportion of assessable data for differential diagnosis, visibility of characteristic ECG structures in cECG, precision of ECG time intervals, and HR. All measurements were conducted in a simulated driving environment. The setup consisted of a segment of a standard vehicle cockpit from a Volkswagen Polo, a car seat equipped with capacitive electrodes, a projector, a computer, and a canvas. It comprised an instrumented car seat, an instrumentation box, a USB safety adapter, and a standard PC running the recording software. The seat was equipped with six small electrodes integrated into the backrest and one large electrode embedded in the seat surface. The electrode in the seat surface was made of conductive textile and measured 27 cm × 34 cm. A small amplifier circuit was located behind each steel plate. The health status of each participant was continuously monitored during the experiment using an additional conventional ECG monitor. The results showed that cECG was inferior in terms of signal quality. It was suitable for assessing strong ECG structures but exhibited limitations for finer structures. For HR, almost all cases in both cECG and rECG could be evaluated. rECG data tended to show slightly higher HR values with larger variations compared to cECG data, but HR remained within the normal range.

In another study, Uguz et al. recorded spikes from an artificial cardiac pacemaker using capacitive ECG (cECG) electrodes, which required no direct skin contact [36]. Using spikes collected from threshold tests, in which the pacing amplitude was lower than normal operation, this study demonstrated the ability of cECG to capture pacemaker spikes and its applicability for monitoring patients with pacemakers. The cECG system employed in this study was based on the car seat presented by Leicht et al. [35], equipped with six cone-shaped stainless steel electrodes. Although the cECG system detected pacemaker spikes less efficiently than the rECG system, it was able to capture the spikes using an adapted algorithm and a simple fusion strategy. Limitations on spike detectability using cECG arose from high noise levels, motion artefacts, and design choices. The measurement setup could be easily extended to other unobtrusive health monitoring applications of cECG, such as chairs or hospital beds.

Mathissen et al. conducted a driving study to induce stress in 24 participants performing various inducement tasks (n-back task, Sing-a-Song Stress Test, and noise exposure) [32]. Each test drive was supervised by a test leader seated in the co-driver seat. Both performance-based and subjective measures were recorded to assess the driver state. The results indicated that the n-back task was the most effective method for inducing stress during driving. Average response times for a simultaneously performed tactile detection response task doubled during the n-back task compared to baseline response times. Sympathetic nervous activation, reflected in increased HR and RR and decreased HR variability, was observed as a physiological reaction to stress-inducing secondary tasks. Subjective ratings indicated low sleepiness levels throughout the approximately one-hour measurement period.

Hu et al. [46] introduced a framework to enhance the detection of driver fatigue from Electroencephalography (EEG) signals. The primary motivation of the study was to address limitations of existing models in capturing both temporal dynamics and spatial dependencies across brain regions. To this end, the authors proposed a spatio-temporal fusion architecture integrating multiple specialized modules. The Recurrent Multi-Scale Convolution Module combined multi-scale convolutions with CNN–Bidirectional Long Short-Term Memory layers and residual connections to model short- and long-term temporal dependencies. For spatial feature extraction, the model employed a dynamic graph convolutional module that encoded inter-electrode relationships by partitioning the EEG channels according to brain regions. The extracted temporal and spatial features were then integrated through a Feature Fusion Module with channel attention, which adaptively weighted the contributions of different modalities prior to classification. The model was evaluated on EEG data collected in real-world driving scenarios. Reported results indicated that the spatio-temporal fusion network with brain region partitioning outperformed several baseline methods. Critical appraisal reveals several strengths. Firstly, the modular design is comprehensive, allowing balanced treatment of temporal and spatial dynamics. The use of a brain-region partitioning strategy improves the biological plausibility of spatial analysis. Nonetheless, several limitations should be noted. The model lacks specialized design for cross-subject EEG classification, resulting in degraded performance in pseudo-online testing. The dynamic graph convolutional module models root-to-region associations but ignores intra-regional node interactions, limiting spatial detail. Furthermore, the computational cost is high, raising concerns regarding the feasibility of real-time, in-vehicle deployment.

Ahlström et al. developed an algorithm that adapts driving time and rest periods during automated driving [29]. The aim was to detect fatigue, relaxation, and sleep. HR and HR variability metrics were used as features to distinguish manual driving from relaxation behind the wheel in a stationary vehicle. The relaxation detection algorithm was trained on a dataset of 100 drivers, evenly distributed across age groups, with half of the participants being female. During the experiment, drivers were instructed to behave as they normally would while steering the car. Data were collected while the vehicle was stationary. The experiment consisted of 40 min driving episodes on different roads conducted during daytime between 09:00 and 16:30. ECG measurements were recorded during the experiment, and heartbeats were extracted by identifying R-peaks. Differentiation between driving and relaxation was achieved using an ML procedure. The dataset was divided into a 70% training set and a 30% test set. Two algorithms were developed: one using subject-dependent HR variability and the other using a subject-independent feature set. The subject-dependent approach achieved superior performance, whereas the subject-independent method was applicable to unknown drivers. Feature selection was performed using Sequential Forward Floating Selection and implemented with a decision tree classifier, employing five-fold cross-validation, ten cross-validation runs, and a trade-off between sensitivity and specificity as the optimization score. The resulting feature subset was used to classify data into relaxation or driving using a Gaussian Support Vector Machine (SVM) classifier. Sleep detection was provided by a camera-based eye-tracking system, with eye closure exceeding 60 s considered indicative of the driver being asleep. Fatigue was estimated by combining the outputs of the relaxation and sleep detection algorithms. The resulting predicted driving time was calculated from the estimated fatigue level, using a Karolinska Sleepiness Scale (KSS) threshold of 7, indicating sleepiness without active effort to stay awake. The experiment yielded promising results for both subject-dependent and subject-independent classification, with accuracies of 88.72% and 80.1%, respectively. It should be noted that these experiments were theoretical, and no empirical evidence on public roads was collected.

Persson et al. investigated the reliability of HR variability (HRV) as a standalone feature for driver sleepiness detection in a real-road driving scenario [30]. Subjective ratings based on the KSS were used as ground truth. Two methodological approaches were employed for HRV analysis. The first was a statistical approach, and the second was an ML approach. Both approaches included a pre-processing stage in which ECG signals were filtered and divided into 5 min epochs. HRV metrics were then calculated for each epoch. The statistical approach involved an analysis of variance, while the ML approach employed a standard ML pipeline to evaluate the suitability of designing a sleepiness classifier based solely on HRV metrics. The dataset comprised data from three separate driver sleepiness experiments, including 18, 24, and 44 drivers. To assess the level of sleepiness, drivers reported their KSS (Karolinska Sleepiness Scale) score every five minutes, which was used as the target value for training the classifiers. Four binary classifiers were employed for classification: K-Nearest Neighbor (kNN), SVM, Adaptive Boosting, and Random Forest. R-peaks were extracted from ECG using the Pan–Tompkins algorithm, and RR time series were calculated as the time intervals between successive heartbeats. The Random Forest classifier achieved the highest accuracy (80.9%), followed by SVM (79.0%), Adaptive Boosting (75.1%), and kNN (67.2%). However, it should be noted that subjective assessments of sleepiness depend on the individual test subjects and may vary between participants.

Finally, Hultman et al. explored the potential of deep neural networks to detect different levels of driver sleepiness based on electrophysiological data [31]. Data were acquired using EEG and electrooculography (EOG). Twelve driver sleepiness experiments were conducted. Five were performed on real roads, including rural roads and highways, while the remainder took place in a driving simulator. Participants were randomly selected from the Swedish register of vehicle owners. Prior to the experiments, drivers were instructed to avoid alcohol for 72 h and to refrain from nicotine and caffeine for three hours. In eleven of the twelve experiments, 225 drivers drove in at least one alert condition during daytime and one sleep-deprived condition at night, having been awake since early morning. In the twelfth experiment, 44 drivers drove only in a sleep-deprived state in the early morning hours after a night shift. The duration of each experiment ranged from 30 to 90 min. Self-reported sleepiness was assessed using the KSS every five minutes, and the KSS values were subsequently used as target values for training the ML algorithm. Sleepiness detection was performed using both binary classification, to distinguish between alert and sleepy states, and regression, to estimate KSS values on a finer scale. The results showed accuracies of 82.0% for classification and 82.6% for regression. EOG data were found to be more informative for driver sleepiness detection than EEG data, although combining EOG and EEG improved performance in some models. As noted previously, subjective assessments of sleepiness depend on individual test subjects and may vary between participants.

There are also other studies focusing on physiological parameter monitoring. For instance, Chou et al. [28] developed a smart wheelchair integrated with an ECG measurement system and wireless transmission capabilities, enabling real-time HR monitoring and remote health assessment for individuals with mobility impairments. Moreover, Chen et al. [47] presented a deep learning model for EEG-based driver fatigue detection. Given that the methodological approaches and findings of these studies largely align with those discussed earlier, they are not analyzed in detail here.

Table 1 summarizes the key information from the aforementioned studies. It provides details on the sensor type, sensor placement, monitored physiological parameters, study objectives, and detection methods, with reference to [3]. Furthermore, the table is divided into three sections. The first section presents sensors that detect parameters via electrical and obtrusive methods. The second section includes vision-based methods for monitoring physiological signals, while the final section lists unobtrusive pressure sensors or PSMs. Quantifying and comparing the results of these studies is challenging due to differences in environmental conditions and measurement methodologies; nevertheless, the findings indicate that these approaches are promising.

Although signals obtained with a PSM may not be explicitly listed in [3], they are still considered part of the physical features. This is because features derived from camera-based facial recordings and those acquired via a PSM through seated interaction differ only minimally. Therefore, in the following section, we focus on existing posture recognition systems that utilise PSMs for data acquisition and ML algorithms for analysis.

## 4. Posture Recognition with Pressure Sensor Mats

Non-obtrusive, continuous monitoring of an individual on a PSM is considered a challenging task [48]. On the one hand, the sensor must meet technical requirements, while on the other hand, it must not obstruct the user who is required to maintain correct posture for a specified duration. We classify all proposed systems for pattern recognition using PSMs according to their respective application fields. Finally, at the end of this section, the presented systems and approaches are summarized in Table 2. The following paragraphs provide details of the objectives, sensors, and analysis methods described in the selected publications, presented in chronological order.

Zemp et al. [22] developed a chair equipped with force and acceleration sensors to assess the accuracy of automatically identifying a user’s sitting posture. They employed five distinct ML methods, SVM, Multinomial Regression, Boosting, Neural Networks (NN), and Random Forest, on the raw data. The study involved 41 participants who were instructed to assume seven different sitting postures (no inclination, forward, backward, left, left leg crossed, right, right leg crossed), repeating each posture four times. Sixteen force sensor values and the backrest angle were used as input features for classification. A Leave-One-Out cross-validation approach was employed to compare the performance of the various classification methods. The Random Forest algorithm demonstrated the best performance, achieving a mean classification accuracy of 90.9% for subjects unknown to the algorithm. Kim et al. also conducted research on a similar topic, analyzing the sitting postures of children between the ages of seven and eleven [48]. Their approach was similar to that of Zemp et al. [22], but they classified only five sitting postures.

Kamiya et al. developed a PSM for the identification of nine different postures [19]. The mat comprised a total of 64 Flexiforce sensors arranged in an 8 × 8 grid. The system unobtrusively recognised postures at a sampling rate of 12.5 Hz. Data were collected from ten male students aged between 21 and 24 years and weighing from 57 kg to 90 kg. Each participant was asked to assume a sequence of nine postures: sitting upright, leaning forward, backward, right, left, sitting upright with right or left leg crossed, and leaning to the right or left with the corresponding leg crossed. Every posture was maintained for two to three seconds, and each participant completed five rounds. The study also evaluated both person-known and person-unknown scenarios, in which training and testing data were derived from the same or different participants, respectively. Classification rates of 98.9 % were achieved for the person-known scenario and 90.6 % for the person-unknown scenario using SVM algorithms with a radial basis function and default parameter values. Classification accuracy increased to 93.9 % when normalization was applied.

Martins et al. developed an intelligent chair capable of detecting and classifying five different postures [37]. A PSM was placed on both the seat and backrest. Posture classification was performed using NN with input images normalized to an interval of [−1; +1], achieving an overall classification accuracy of 98.1% at a sampling rate of 18.4 Hz. The system employed eight air cushions (four in the seat and four in the backrest), each coupled with a pressure sensor to measure internal pressure. Various parameter combinations were tested for the classification algorithms, including the number of neurons (10 to 35 in steps of five), number of layers (one, two, and three), transfer functions (Tansig and Logsig), and network training functions (Levenberg–Marquardt algorithm, Scaled Conjugate Gradient algorithm, and Resilient Backpropagation algorithm). The optimal configuration consisted of a single-layer network with 15 neurons, Tansig as the transfer function, and the Resilient Backpropagation algorithm. Honeywell 24PC Series piezoelectric gauge pressure sensors were used to measure the pressure in each air cushion. Experiments were conducted on 15 male and 15 female participants with a mean age of 20.9 years, mean weight of 67.8 kg, and mean height of 1.72 m. The postures examined were leaning forward, backward, left, right, and sitting upright. A total of 29 participants were used to train the network, while one participant was used for testing.

Ma et al. focused on a wheelchair assist system for mobility-impaired individuals capable of recognising postures using a cushion-based system [38]. Pressure sensors were employed to capture data from the user’s postures. Posture classification was performed using the J48 classifier (pruning confidence C = 0.25 and minimum number of instances per leaf M = 2), which achieved the highest accuracy of 99.48% among five supervised classification techniques, at a sampling rate of 2 Hz. Twelve FSR 406 pressure sensors were integrated into the wheelchair, with seven placed in the seat and five in the backrest.

Roh et al. developed a sitting posture monitoring system based on a low-cost load cell array and ML techniques [39]. The seat of a chair was replaced by a custom-made seat comprising four load cells. Six postures were monitored, and seven classifiers were evaluated. The results indicated that a SVM algorithm with a radial basis function kernel was most suitable for classifying sitting postures using sensors exclusively on the seat plate. The kernel achieved an average classification accuracy of 97.20% at a sampling rate of 1 Hz. Tests were conducted on 24 male participants with a mean age of 27.6 years, mean height of 1.74 m, and mean weight of 71.9 kg. The sitting postures assessed were upright sitting with backrest, upright sitting without backrest, forward leaning with backrest, forward leaning without backrest, left sitting, and right sitting.

Rosero-Montalvo et al. developed a system to inform wheelchair users about correct sitting posture [18]. The system is based on a pressure sensor network embedded in the seat and back cushions of the wheelchair. Three pressure sensors were integrated into the seat, and one ultrasonic sensor was placed on the backrest to determine the distance between the user’s back and the wheelchair backrest. Four postures (sitting upright, leaning forward, left, and right) were measured on five participants, with each posture repeated 25 times per participant. Classification was performed using kNN (k = 1). Experimental results showed an average classification accuracy of 75% using ten-fold cross-validation. Additionally, the system reduced the amount of required training data by 88%.

Diao et al. proposed a low-cost, real-time smart mat system for sleep posture recognition based on frequency channel selection [40]. The system unobtrusively recognizes postures using a dense flexible sensor array comprising 64 × 64 sensors. The mat consists of a five-layer structure that acquires the pressure distribution and converts it into voltage signals. A data acquisition circuit connected to the mat is employed for signal control and data collection. The system captures the voltage from each sensor, processes the pressure distribution image, and subsequently classifies the sleep posture. Measurements were performed on 21 participants. A Convolutional Neural Network (CNN) with one convolutional layer, four bottleneck blocks, and one dense layer was used for classification. Training was conducted using Leave-One-Person-Out Cross-Validation. Prior to training, the data were pre-processed using threshold filtering, Gaussian filtering, and adjacent affected noise removal, and transformed via two-dimensional Discrete Cosine Transformation. Experimental results demonstrated an accuracy of up to 95.43% for short-term experiments and up to 86.80% for overnight experiments.

Yuan et al. [41] introduced a velostat-based PSM that captures human body pressure distributions as image streams. Using deep NN, it classifies 4 sleeping postures (98.8% accuracy) and 13 postures (96.6% accuracy) while performing dynamic activities. The system stands out for its low cost, portability, and robust performance. However, several limitations need to be considered. Only 14 participants were included, predominantly male. This restricts the generalizability of the results, especially for different body types, ages, and health conditions. Although 30,000 samples were collected, they stem from a small participant group. The apparent high accuracy may partly result from overfitting to limited subject variability. Also, experiments appear to have been conducted under laboratory conditions. Real-world variability could affect robustness. Finally, the study compares three deep learning architectures but does not benchmark against simpler models (e.g., SVM, Random Forests) that might achieve comparable performance with lower computational costs.

In the selected studies, the most important parameters for posture recognition using a PSM—namely sensor placement, number of sensors, recognized postures, classification techniques, and algorithmic accuracy—are summarized in Table 2. It can be observed that the number of sensors is relatively low when using pressure sensors on a seat (≤50) and substantially higher when a PSM is integrated into a mattress (≥51). All studies employing pressure sensors recognized at least the four classical postures: no inclination, forward inclination, left inclination, and right inclination, with additional postures detected depending on the specific study. Various algorithms were applied for posture classification, achieving accuracies ranging from 75.0% to nearly 100.0%. The suitability of different algorithms depends on the particular application context.

Several other studies have also explored sitting/sleeping posture classification using PSMs and ML. For instance, Wu et al. [49] classified four sleep postures in real-time on PSMs of varying sensor densities using NNs, while Green et al. [50] proposed a system to classify sleeping postures and sleep versus awake states depending on temporal CNN. Moreover, Elsharif et al. [51] applied clustering to PSM data to automatically identify and track limb positions across common sleep postures. Further interesting, noteworthy research works were published by Ren et al. [52] and Tsai et al. [53]. Ren et al. developed a system that recognized 20 different postures of 32 people lying in beds (16 males and 16 females) using a Kinect sensor and a fuzzy logic and ML combined algorithm. They intended to allow an automated healthcare system to recognize the body-limb posture of a patient lying in bed for automated healthcare service. Tsai et al. proposed a sitting posture recognition system based on a PSM comprised of 13 pressure sensors. They employed five ML algorithms and compared the results for ten sitting postures. A total of six subjects participated in the study, consisting of three males and three females. Since the methodological approaches and findings of these works are largely in line with the studies discussed above, they are not examined here in detail.

Table 2 provides an overview of studies that employed ML algorithms for posture recognition using a PSM. It summarizes information on sensor placement and number, recognized postures, applied classification techniques, and achieved accuracies.

## 5. Discussion

The advances in technical innovations have opened significant research avenues in personal monitoring, with applications extending to automotive safety and beyond. Consequently, methods for monitoring posture and physiological parameters have evolved considerably in recent years. This review aimed to provide a comprehensive overview of existing techniques for monitoring posture and physiological parameters of individuals using a PSM. Wired monitoring is increasingly considered suboptimal due to its obtrusiveness and the impracticality of continuous long-term monitoring. Therefore, it is essential for monitoring systems to be unobtrusive and non-intrusive. Nevertheless, continuous unobtrusive monitoring on a PSM remains a challenging task. With the enhanced performance of modern sensors and increased computational capacity for data analysis, however, the integration of embedded systems into vehicles has become increasingly feasible. Achieving high classification accuracy depends critically on the technical specifications of the sensors, the quality of the datasets, and the optimal application of ML algorithms.

While most studies on posture recognition using PSMs originate from healthcare or office contexts, their findings provide valuable insights that could be adapted for vehicle drivers. The fundamental principles of PSM-based posture detection—such as sensor placement, pressure distribution analysis, and ML-based classification—are broadly applicable to any seated human subject. For example, algorithms developed to recognize slouching, leaning, or other posture changes in office chairs or wheelchairs could inform the detection of unsafe or fatigue-related postures in drivers. However, several limitations must be considered when translating these results into the driving environment. Unlike static or semi-static contexts, vehicle operation introduces dynamic movements, vibrations, and rapid postural adjustments due to steering, acceleration, and braking. These factors may interfere with pressure patterns and reduce classification accuracy. Furthermore, the range of postures relevant to driving is narrower than in hospital or office settings; subtle forward or lateral shifts may be more critical than general posture variations. Driver monitoring also often requires real-time detection to prevent potentially life-threatening situations, whereas many healthcare or office applications are designed for post hoc analysis.

During our research for this review, we identified a previous study presenting a state-of-the-art analysis specifically focused on detecting sitting positions and habits [54]. Our approach differs from this prior work by adopting a broader analytical scope. In our study, posture detection using pressure sensors was investigated, encompassing both minimal sensor setups and PSMs with over a thousand sensors. Furthermore, we extended the analysis to include the detection of physiological parameters such as fatigue and HR, incorporating a variety of sensor types including electrical, mechanical, and visual sensors. In contrast, the earlier study primarily concentrated solely on posture detection, without addressing physiological parameter monitoring. Given the limited number of studies investigating posture recognition using PSMs in seated individuals, studies involving bedridden participants were also considered to provide additional insights into sensor placement, pressure patterns, and classification methodologies.

Movements on a seat are by no means static, but rather highly dynamic. Most existing solutions for posture recognition primarily focus on classifying predefined extreme positions, such as a full tilt to one side. However, such extreme postures occur only in rare cases. In most instances, individuals maintain positions between these extremes, yet there appears to be a paucity of studies addressing intermediate postures. Additionally, some previous studies exhibit further limitations. In [16,19,40,51], a high number of sensors were used in the PSM, rendering the solutions relatively costly. Another limitation is that in [19,39], where measurements were conducted exclusively on male subjects. Given the differences in body structure between sexes, this may result in distinct pressure patterns on the mat. To ensure representativeness, both sexes should be included in balanced numbers. Furthermore, refs. [16,18,19,38] conducted measurements on only five to twelve participants, which constitutes a very limited sample size. It is also noteworthy that Rosero-Montalvo et al. [18] achieved an accuracy of 75.0%, substantially lower than the values reported in other studies, which exceed 95.0%.

In addition to methodological considerations, practical challenges must be considered when implementing PSMs in vehicle seats. These include:**Durability:** Vehicle seats are subject to repeated loading, temperature variations, and vibrations, which may affect the longevity and reliability of the sensors.**Cost:** Integrating high-resolution pressure mats into mass-produced vehicles could be expensive, which may limit commercial adoption.**Integration:** Sensors must be seamlessly incorporated into existing seat designs without reducing comfort, interfering with safety systems (e.g., airbags), or affecting seat ergonomics.

Addressing these challenges is critical for real-world deployment. While research studies provide proof-of-concept results, additional work is needed to adapt PSM-based systems to the dynamic and demanding environment of automotive seats.

Furthermore, a notable limitation exists in studies attempting to detect fatigue through physiological measurements such as HR and RR using a PSM. Many of these studies [29,30,32] rely on the KSS for sleepiness assessment. While the KSS is simple and widely adopted, it has several limitations: it depends on participants’ self-perception, rendering it susceptible to bias; it can be influenced by situational factors such as stress or ongoing activities; and it provides only a snapshot of sleepiness, offering no information on long-term sleep deprivation or circadian effects. Consequently, identical conditions may yield differing results. Another limitation in [29,30,31,32,35,36] is that the measurements were conducted obtrusively, which inherently creates discomfort for the participants.

Nakane et al. [17] demonstrated that fatigue may be detectable through the use of a sensor-equipped chair. However, this study exhibits several limitations. The sample size was small, and the evaluation function applied to the centre of pressure data does not converge to a single point. Small experimental groups entail multiple disadvantages. Firstly, they limit the statistical power of the study, increasing the likelihood of errors, meaning that real effects may remain undetected. Secondly, findings from a small sample may not generalize well to the broader population, reducing the external validity of the results. Thirdly, individual differences or outliers can disproportionately influence the outcomes, making the results less stable and reliable. Other research groups may encounter similar limitations due to practical constraints inherent to small sample sizes. Conducting large-scale measurement campaigns requires considerable time, meticulous preparation, and participant familiarization. In our own research, completing data collection took approximately twelve months, a timeframe that is often unavailable to other groups, which further explains the reliance on smaller experimental cohorts in prior studies.

Furthermore, the studies analyzed in this work emphasise minimizing the number of sensors used in the seat due to the associated high costs, which render extensive sensor setups impractical and economically unfeasible. Employing fewer sensors offers several practical advantages, including simpler installation and reduced maintenance requirements, as there are fewer components to manage and troubleshoot. Additionally, a minimal sensor setup occupies less physical space, which is particularly beneficial in confined environments such as vehicle seats. The reduced number of sensors also decreases the volume of data generated, thereby simplifying storage, processing, and analysis. However, reliance on a limited number of sensors introduces significant constraints. A smaller sensor array captures less information, potentially compromising the accuracy of detecting subtle physiological or behavioral states. Moreover, fault tolerance is reduced, as the failure of a single sensor may result in the loss of critical data. Overall reliability and precision are thus affected, complicating the detection of mild or early-stage conditions. Systems with minimal sensors often struggle to account for individual differences, capturing general trends rather than personalized patterns. Finally, such setups offer limited flexibility, making them less adaptable to diverse scenarios or applications.

The camera used by Lyra et al. [12] yields promising results; however, data security is not guaranteed, and private information could be misused. Ensuring robust data security in camera-based systems offers several advantages. Primarily, it protects user privacy by safeguarding sensitive personal information from unauthorized access. Furthermore, strong security measures help maintain user trust, as individuals are more likely to engage with systems when they are confident their data are protected. Effective data protection also mitigates the risk of misuse, preventing information from being exploited for surveillance, profiling, or other unintended purposes. Conversely, inadequate data security introduces significant risks. Unauthorized access may result in data leaks, potentially leading to identity theft or financial harm. Breaches of personal information undermine user trust and can damage the reputation of the system or provider. Legal consequences may also arise, including fines, compensation claims, and regulatory penalties. Moreover, unsecured data remain vulnerable to misuse for surveillance, profiling, or other malicious purposes. Finally, insufficient data protection may limit the practical implementation of systems or hinder research, as ethical and legal requirements could restrict the use of sensitive data.

Features and functions described in Section 3 and Section 4 encompass monitoring a person’s posture, thereby enabling inferences regarding the applied pressure on the PSM, as well as measuring physiological parameters for detecting potentially life-threatening situations, particularly HR and RR. Early detection of these parameters can indicate health emergencies, facilitating timely interventions such as triggering an emergency call, for instance if a patient experiences a cardiac event in a hospital or while driving. Achieving reliable detection requires a sufficiently comprehensive database, which can be obtained by optimizing the number and placement of PSM/pressure sensors. Increasing the number of sensors enhances algorithmic accuracy; however, Wu et al. [49] demonstrated that it is possible to reduce the number of sensors while maintaining satisfactory performance. Beyond a certain threshold, additional sensors no longer significantly influence results. Maintaining an appropriate balance between sensor quantity and the range of classified postures is therefore essential. Different postures necessitate heightened sensitivity to pressure variations on the PSM. As shown in Table 2, improved accuracy was observed in scenarios involving postures beyond the classic extremes (forward, backward, left, right), with studies recognizing eight or nine postures consistently employing at least 64 sensors [16,40,51].

Direct comparison across all studies remains challenging due to variations in environmental conditions; nonetheless, the consistently reported results—predominantly exceeding 95.0%—are considered satisfactory. Similar considerations apply to the measurement of physiological parameters, where differences in sensor type and placement contribute to variability among studies. Moreover, the distinction between obtrusive and unobtrusive methods is significant. Obtrusive techniques for driver monitoring offer the advantage of providing precise and detailed physiological data. Direct-contact sensors, such as ECG, EEG, or skin conductance devices, enable continuous measurement of HR, RR, eye movements, and muscle activity, allowing for early detection of fatigue or stress before critical thresholds are reached. This precision and continuity make obtrusive methods particularly effective in capturing subtle changes in the driver’s physiological state.

However, these methods also have substantial drawbacks. They may cause discomfort, as wearables or electrodes can restrict movement or irritate the skin. Hygiene and maintenance present additional challenges, requiring regular cleaning and periodic replacement of sensors. User acceptance is often limited, as many individuals are reluctant to employ invasive or physically intrusive devices. Installation and calibration can be time-consuming and error-prone, while high-quality physiological sensors entail considerable costs, limiting feasibility for widespread deployment. Furthermore, subjective approaches, such as assessments by a co-driver, introduce bias. In contrast, this review highlighted the effectiveness of unobtrusive measurement methods, particularly the use of a PSM, in capturing the desired information.

As demonstrated, posture recognition using a PSM in combination with ML algorithms can achieve accuracies approaching 100%, whereas measuring physiological parameters with the same system remains considerably more complex. The diversity of the data to be collected can be leveraged both in processing and in practical implementation when deploying an integrated system. Specifically, based on the approaches discussed above, the system hardware should incorporate both PSMs and sensors capable of monitoring physiological parameters such as HR, RR, and fatigue, as well as the user’s weight and areas of sustained pressure on the body. Although solutions addressing parts of this problem already exist, none provide a comprehensive solution in their entirety.

## 6. Conclusions

Preventing road traffic accidents through unobtrusive and continuous monitoring of the driver’s posture and physiological parameters remains a considerable challenge. Over recent years, numerous studies have focused on posture recognition using PSMs, with applications ranging from continuous physiological monitoring to pressure distribution analysis. In this review, we provided an overview of these studies and their findings. While many of the approaches reviewed demonstrate promising results using diverse techniques, none specifically address the detection of mundane or dynamic movements. Consequently, the need for systems capable of continuously and unobtrusively monitoring drivers, while actively mitigating accident risk, remains unmet.

Our research group is currently addressing this gap by recognizing both static sitting postures and dynamic movements on a PSM using ML techniques. To ensure diversity and minimize bias, data are collected from a large participant cohort with a balanced male-to-female ratio. For static postures, we extend existing classification methods to a larger set of postures, enabling a more detailed understanding of driver behavior. For dynamic movements, we treat the problem as a regression task to capture time-dependent patterns. Given the high volume of temporal data, advanced signal processing techniques are employed to reduce dimensionality while preserving relevant information and optimizing model performance. Additionally, we investigate whether models generalize to new participants or require individual calibration, thereby providing insights into real-world applicability.

Future research should prioritize several key directions. Firstly, optimizing the number and placement of sensors, alongside increasing sensitivity to subtle deviations in posture or physiological signals, is essential for improving data quality. Secondly, continuous monitoring over time is necessary, as drivers rarely remain perfectly still, and capturing temporal dynamics is crucial for detailed and reliable assessments. Thirdly, careful consideration of sensor physics, placement, and system topology is required, alongside the application of advanced signal processing and data reduction techniques to manage the growing data volume. Fourthly, the development and evaluation of ML algorithms capable of handling high-dimensional, time-dependent data while remaining generalizable across individuals remains an open challenge. Finally, data security and privacy must be ensured to prevent the misuse of sensitive personal information.

Addressing these technical challenges—low signal-to-noise ratios, dynamic driver behavior, system calibration, real-time processing, and privacy protection—is essential for developing practical, robust, and widely deployable unobtrusive driver monitoring systems. By combining optimized sensor configurations with sophisticated signal processing and ML pipelines, future studies can progress towards safer and more effective in-vehicle monitoring solutions.

## Figures and Tables

**Figure 1 sensors-25-06238-f001:**
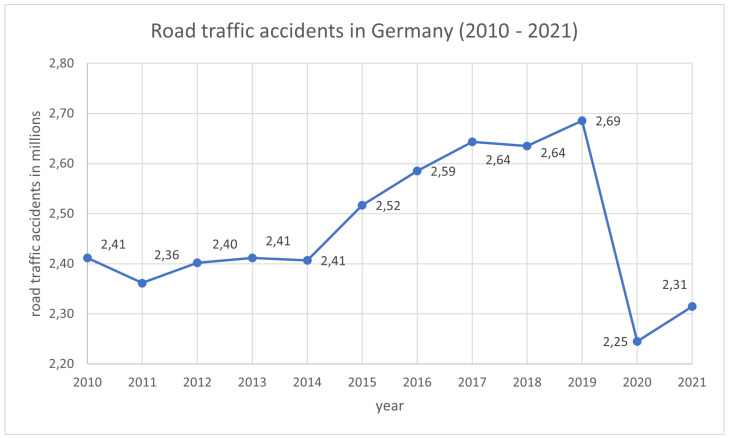
Annual development of road traffic accidents in Germany from 2010 to 2021 [1].

**Figure 2 sensors-25-06238-f002:**
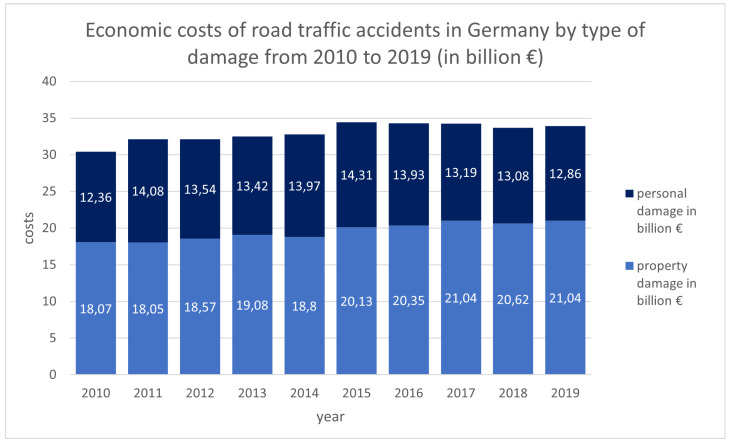
Economic costs of road traffic accidents in Germany by type of damage from 2010 to 2019 (in billion EUR); light blue: costs of property damage, dark blue: costs of personal damage [1].

**Figure 3 sensors-25-06238-f003:**
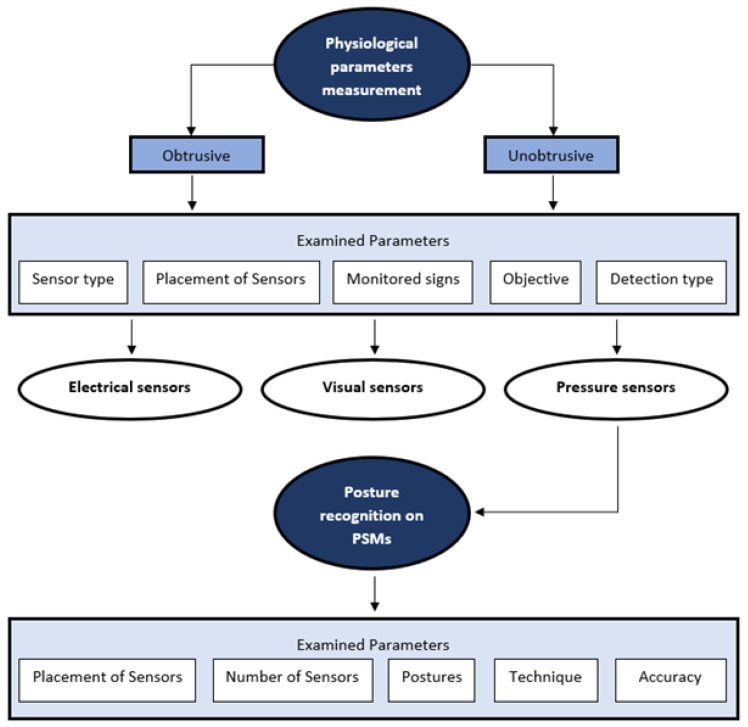
Structure of this review paper.

**Figure 4 sensors-25-06238-f004:**
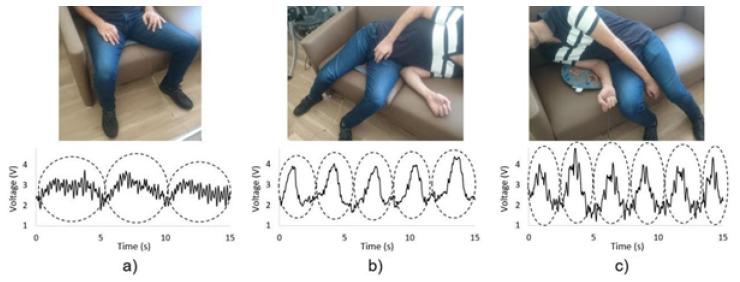
BCG signals of a person (**a**) sitting on the pressure sensing mat with no inclination, (**b**) seating on the pressure sensing mat but leaning away from it, and (**c**) seating next to the pressure sensing mat but leaning over it [27].

**Table 1 sensors-25-06238-t001:** Overview of literature that has dealt with physiological parameters measurement (HR: Heart rate; BP: Blood pressure; RR: Respiration rate; BT: Body temperature).

Author	Sensor Type	Placement of the Sensors	Monitored Signs	Objective	Detection Type
Lima et al. [27]	Strain-Gauge sensor (1) Ballistocardiography (2)	Top end of the seat (1) Wired to the driver (2)	HR and RR	Physiological parameters monitoring	Biological features
Hu et al. [28]	Electroencephalography	Head of the driver	Fatigue	Fatigue detection	Biological features
Ahlström et al. [29]	Electrocardiography (1) Camera (2)	Wired to the driver (1) In the driver’s cab (2)	HR, Fatigue	Fatigue detection	Hybrid features (Biological and Physical)
Persson et al. [30]	Electroencephalography	Wired to the driver	HR	Sleepiness detection	Biological features
Hultman et al. [31]	Electroencephalography, Electrooculography	Wired to the driver	HR and eye motion	Fatigue detection	Biological features
Lyra et al. [12]	Camera	Camera above the patient	HR, BP, RR and BT	Physiological parameters monitoring	Physical features
Mathissen et al. [32]	Co-driver	A test leader sitting next to the driver	HR and RR	Sleepiness detection	Subjective
Nakane et al. [17]	Pressure sensing mat	16 × 16 Matrix	Centre of pressure	Postural sway detection	Physical features
Tu et al. [33]	Pressure sensing mat	32 × 32 Matrix	Pressure distribution	Fatigue detection	Physical features
Huang et al. [34]	Pressure sensing mat	42 × 48 Matrix under the patient	HR and RR	HR and RR estimation	Physical features
Leicht et al. [35]	Pressure sensors	1 pressure sensor in the seat; 6 pressure sensors in the backrest	HR and RR	Physiological parameters monitoring	Physical features
Uguz et al. [36]	Pressure sensors	1 pressure sensor in the seat; 6 pressure sensors in the backrest	HR and RR	Physiological parameters monitoring	Physical features

**Table 2 sensors-25-06238-t002:** Overview of literature that has dealt with posture recognition on pressure sensing mats using Machine Learning.

Author	Placement of the Sensors	Number of Sensors	Postures Recognised	Classification Techniques	Accuracy
Rosero-Montalvo et al. [18]	3 pressure sensors in the seat; 1 ultrasonic sensor in the backrest	4	4 (no inclination, forward, left, right)	k-Nearest Neighbors	75%
Kamiya et al. [19]	8 × 8 Matrix on the seat	64	9 (no inclination, forward, backward, left, left leg crossed, leaning left with left leg crossed, right, right leg crossed, leaning right with right leg crossed	Support Vector Machines	98.9%
Zemp et al. [22]	10 pressure sensors in the seat; 4 pressure sensors in the backrest; 2 pressure sensors in the armrests	16	7 (no inclination, forward, backward, left, left leg crossed, right, right leg crossed)	Support Vector Machines, Multinomial Regression, Boosting, Neural Networks, Random Forest	≤90.9%
Tu et al. [33]	32 × 32 Matrix on the seat	1024	No posture, but fatigue	7 different models, but Random Forest was best	92.0%
Martins et al. [37]	4 pressure sensors in the seat; 4 pressure sensors in the backrest	8	5 (no inclination, forward, backward, left, right)	Artificial Neural Network	98.1%
Ma et al. [38]	7 pressure sensors in the seat; 5 pressure sensors in the backrest	12	5 (no inclination, forward, backward, left, right)	J48 Classification	99.5%
Roh et al. [39]	4 load cells in the seat	4	6 (no inclination, forward, forward with offset, backward, left, right)	7 different models, but Support Vector Machine was best	97.2%
Diao et al. [40]	32 × 32 Matrix on the mattress	1024	4 (supine, prone, left, right)	Support Vector Machines, k-Nearest Neighbors	95.0%
Yuan et al. [41]	30 × 29 (Not from paper)	870	4 (supine, prone, left, right) 13 (dynamic activities)	Deep Neural Networks	99.3% sleeping postures 96.6% dynamic activities

## Data Availability

No new data were created or analyzed in this study. Data sharing is not applicable to this article.

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
