# Peer review of "Analysis of Physiological Parameters and Driver Posture for Prevention of Road Accidents: A Review"

_sensors, 2025, doi:10.3390/s25196238_

Round 1
Reviewer 1 Report
Comments and Suggestions for Authors
The manuscript presents a general overview of the state of the art of methods for detecting physiological parameters and posture, invasive and non-invasive, focusing on conduction. In conclusion, several research opportunities were identified, such as the quantity, quality, and interpretability of the results. It is a well-structured manuscript. However, the relationship between this combination of two approaches and the risk of accident has not been thoroughly discussed.
General concept comments
• The review is clear and relevant to the field. However, it appears more as an overview of the methods, and a deeper analysis of the relationship between the results and the risk of accident should be highlighted.
• Even though it is not a systematic review, it is highly recommended to clarify the queries and the search sequences to identify the focus and completeness of the search as well as promote the reproducibility.

Author Response
Dear Reviewer,
Thank you very much for taking the time to carefully review our manuscript and for providing thoughtful comments and suggestions. We appreciate your constructive feedback, which has helped us to improve the clarity, rigor, and overall quality of our work. Below, we provide a detailed point-by-point response addressing each of your comments.
Comment 1:
This review is clear and relevant to the field. However, it appears more as an overview of the methods, and a deeper analysis of the relationship between the results and the risk of accident should be highlighted.
Response 1:
We sincerely thank the reviewer for this valuable comment. We agree that the relationship between the results and accident risk is an important aspect of safety-related research. However, the primary focus of our work is the detection of physiological parameters and posture recognition rather than accidents themselves. The discussion of accidents in our manuscript was intended to provide motivation and context for the relevance of timely detection systems. We have clarified this distinction in the introduction (Lines 60-72) to better guide the reader’s understanding of the scope of our work.
Comment 2:
Even though it is not a systematic review, it is highly recommended to clarify the queries and the search sequences in order to identify the focus and completeness of the search as well as promote the reproducibility.
Response 2:
We fully agree with the reviewer that transparency regarding the search process is essential to clarify the focus and promote reproducibility. Therefore, we have now added a detailed description of our search strategy, including the selection procedure, to the methodology section (Lines 116–121).
Comment 3:
Table 1 and 2 descriptions should include a mention of invasive or non-invasive condition because it is a relevant characteristic discussed in the text.
Response 3:
We thank the reviewer for this valuable suggestion. In our work, we distinguish between obtrusive and unobtrusive methods. To clarify this further, here are some examples:
– Invasive methods involve sensors placed inside the body (e.g., implanted sensors).
– Obtrusive but non-invasive methods disturb the driver somehow, for example ECG electrodes, which need direct contact, but are not invasive.
– Unobtrusive methods do not require direct contact with the body and they are inconspicuous like pressure sensing mats.
All papers we added in our work are non-invasive.
Comment 4:
Besides, table 1 should be considered a column with the processing methods and results in order to be complete and be comparable with table 2.
Response 4:
We thank the reviewer for this helpful comment. The purpose of Table 1, however, is to provide an overview of the different methods used for monitoring physiological parameters. Since these studies rely on heterogeneous approaches, the results are not directly comparable. In contrast, Table 2 summarizes studies that share a common methodological basis (posture/state recognition combined with machine learning), which allows a meaningful comparison of their results. We have revised the captions of both tables to better reflect their respective objectives and to clarify this distinction for the reader.
Comment 5:
References are relevant, but it is highly recommended to include more journal references instead of conference papers to improve to confidence and completeness of the work.
Response 5:
We fully agree with the reviewer’s recommendation. Following this suggestion, we performed an additional literature search with a particular focus on recent journal publications. As a result, we have extended our reference list by six additional journal papers, which further strengthen the confidence and completeness of our work and highlighted them in the paper.
Reviewer 2 Report
Comments and Suggestions for Authors
This manuscript presents a comprehensive overview of physiological parameter monitoring and posture recognition methods for applications in driver fatigue detection and accident prevention. The paper is well-structured, and clearly compares different methods. It systematically categorizes previous works by sensor types, monitored parameters, and recognition algorithms, as well as in highlighting the underexplored area of PSM-based fatigue detection. However, there are still some areas that need improvement before the publication.
- While the review framework is being about accident prevention in drivers, a substantial part of studies involves non-driving contexts (e.g., bedridden or office settings). A clear discussion on the transferability of these results to driving environments is needed.
- The review tends to describe methods without sufficiently evaluating their advantages, limitations, and potential integration into real applications. A more comprehensive synthesis would strengthen the conclusions
- There is limited discussion on real-world feasibility, such as durability, cost, and integration challenges of PSMs in automotive seats. Including this would improve practical relevance.
- The paper points out the lack of research on detecting seat movements but does not suggest possible solutions; it would be valuable to briefly discuss potential sensing or algorithmic approaches to address this gap.
- In addition to summarizing sensing and recognition methods, the review could be enhanced by including driver state monitoring with adaptive control implementation. For example, recent work on “A Human-Machine Shared Control Framework Considering Time-Varying Driver Characteristics, IEEE Transactions on Intelligent Vehicles, vol. 8, no. 7, pp. 3826-3838, July 2023” offers relevant insights on how monitored physiological and behavior data can be directly utilized to improve road safety.
Author Response
Dear Reviewer,
Thank you very much for taking the time to carefully review our manuscript and for providing thoughtful comments and suggestions. We appreciate your constructive feedback, which has helped us to improve the clarity, rigor, and overall quality of our work. Below, we provide a detailed point-by-point response addressing each of your comments.
Comment 1:
While the review framework is being about accident prevention in drivers, a substantial part of studies involves non-driving contexts (e.g., bedridden or office settings). A clear discussion on the transferability of these results to driving environments is needed.
Response 1:
We thank the reviewer for this important remark. We have now added a dedicated paragraph to the discussion section (Lines 546-560) that explicitly addresses the transferability of results from non-driving contexts (e.g., medical or office environments) to driving scenarios. This addition clarifies the potential and the limitations of applying such findings to driver monitoring and accident prevention.
Comment 2:
The review tends to describe methods without sufficiently evaluating their advantages, limitations, and potential integration into real applications. A more comprehensive synthesis would strengthen the conclusions
Response 2:
We appreciate this valuable comment. Many of the reviewed methods share similar strengths and limitations, which is why we opted to present a general evaluation in the discussion section rather than repeating similar points for each individual method. We have now revised and slightly expanded the discussion (Lines 596–637) to emphasize these commonalities more clearly and to address the potential for integration into real-world applications in a concise and comprehensive manner.
Comment 3:
There is limited discussion on real-world feasibility, such as durability, cost, and integration challenges of PSMs in automotive seats. Including this would improve practical relevance.
Response 3:
We thank the reviewer for this valuable suggestion. We have added a dedicated paragraph in the discussion section (Lines 573–595) that addresses the real-world feasibility of PSMs, including aspects such as durability, cost considerations, and integration challenges in automotive seats. This addition enhances the practical relevance of our work and highlights important factors for future implementation.
Comment 4:
The paper points out the lack of research on detecting seat movements but does not suggest possible solutions; it would be valuable to briefly discuss potential sensing or algorithmic approaches to address this gap.
Response 4:
We thank the reviewer for this helpful suggestion. Following this comment, we have added a short paragraph to the introduction section (Lines 101–107) in which we outline potential approaches to address this research gap. In particular, we briefly describe the algorithmic strategy we are currently investigating in our ongoing project, which involves a bigger dataset, dimension reduction techniques, feature extraction and new machine learning approaches. This addition provides the reader with a perspective on possible solutions and future research directions.
Comment 5:
In addition to summarizing sensing and recognition methods, the review could be enhanced by including driver state monitoring with adaptive control implementation. For example, recent work on “A Human-Machine Shared Control Framework Considering Time-Varying Driver Characteristics, IEEE Transactions on Intelligent Vehicles, vol. 8, no. 7, pp. 3826-3838, July 2023” offers relevant insights on how monitored physiological and behaviour data can be directly utilized to improve road safety.
Response 5:
We thank the reviewer for this excellent suggestion and for pointing us to this relevant reference. We have added this work to our review and briefly discussed its contribution in the introduction section (Lines 87–91). However, we would like to emphasize that the focus of our review is on the monitoring and recognition of the driver’s state rather than the control strategies for the vehicle. Nevertheless, we agree that the connection between driver monitoring and adaptive vehicle control is highly relevant for improving road safety, and we highlight this as an important direction for future research.
Reviewer 3 Report
Comments and Suggestions for Authors
In this review article, the authors present an overview of existing accident prevention methods that monitor a person’s physiological state, movements, and physiological parameters. Please see below my specific comments that need to be addressed:
1. The authors need to elaborate on the rationale for conducting this review.
2. While the authors have reported existing literature and findings in the field, the review currently reads more like a narration than a critical analysis. I would like to see the authors' commentary on each work cited - highlighting the strengths, limitations, and unique contributions of each.
3. In addition to the discussion section at the end, a critical evaluation of each referenced paper should be provided throughout the review.
4. Based on the reviewed literature, what are the authors' recommendations for the reader? A scientific critical appraisal is missing - what is strong, weak, or lacking in the current body of research? What are the remaining challenges and future directions in the field? These elements need to be clearly addressed.
Comments on the Quality of English LanguageEnglish could be improved.
Author Response
Dear Reviewer,
Thank you very much for taking the time to carefully review our manuscript and for providing thoughtful comments and suggestions. We appreciate your constructive feedback, which has helped us to improve the clarity, rigor, and overall quality of our work. Below, we provide a detailed point-by-point response addressing each of your comments.
Comment 1:
The authors need to elaborate on the rationale for conducting this review.
Response 1:
We thank the reviewer for this helpful comment. We have revised the introduction (Lines 92–100) to elaborate on the rationale for conducting this review, clearly outlining the motivation, relevance, and intended contribution of our work to the field.
Comment 2:
While the authors have reported existing literature and findings in the field, the review currently reads more like a narration than a critical analysis. I would like to see the authors' commentary on each work cited - highlighting the strengths, limitations, and unique contributions of each.
In addition to the discussion section at the end, a critical evaluation of each referenced paper should be provided throughout the review.
Response 2:
We appreciate this valuable comment. Many of the reviewed methods share similar strengths and limitations, which is why we opted to present a general evaluation in the discussion section rather than repeating similar points for each individual method. We have now revised and slightly expanded the discussion (Lines 573–637) to emphasize these commonalities more clearly and to address the potential for integration into real-world applications in a concise and comprehensive manner.
Comment 3:
Based on the reviewed literature, what are the authors' recommendations for the reader? A scientific critical appraisal is missing - what is strong, weak, or lacking in the current body of research? What are the remaining challenges and future directions in the field? These elements need to be clearly addressed.
Response 3:
We thank the reviewer for this important remark. We have revised the conclusion section to include a concise critical appraisal of the current state of research, highlighting its main strengths, weaknesses, and existing gaps. In addition, we now provide clear recommendations for future research directions and emphasize the key challenges that need to be addressed to advance the field (Lines 717–731).
Reviewer 4 Report
Comments and Suggestions for Authors
This article presents a literature review on physiological parameter monitoring and posture recognition using pressure sensing mats (PSM), with a focus on applications for road accident prevention. The authors analyze recent research involving unobtrusive sensors and machine learning algorithms to detect fatigue, heart rate, respiration rate, and body posture. The paper aims to identify research gaps and highlight the potential of PSM-based systems as non-invasive tools for improving driver safety.
Below are several points that, in my opinion, require revision or improvement:
1. I don’t see a clear explanation of how the literature was selected. There’s no mention of which databases were used or how many papers were initially found.
2. Although the focus is supposed to be on drivers, many examples come from hospital, office, or wheelchair contexts. This weakens the relevance for the transportation domain.
3. Table 1 contain a large amount of valuable information, but they are dense and hard to interpret at a glance. I recommend improving their readability by applying clearer formatting, grouping similar entries, or highlighting key values. This would make the comparisons more accessible to the reader.
4. Several studies rely on self-reported fatigue levels (e.g., KSS), which are not reliable. I’d expect stronger critique of this limitation.
5. The discussion mostly summarizes findings without analyzing the reasons for limitations, low accuracy, or practical challenges in real-world use.
6. The conclusion is too general. I’d prefer to see a more structured suggestion of next research steps or open technical problems.
The overall level of English in the article is understandable but uneven. Most technical content is conveyed clearly, but there are noticeable issues with grammar, phrasing, and style throughout the text. While the message is generally intelligible, the manuscript would benefit from professional proofreading to improve fluency and polish.
Author Response
Dear Reviewer,
Thank you very much for taking the time to carefully review our manuscript and for providing thoughtful comments and suggestions. We appreciate your constructive feedback, which has helped us to improve the clarity, rigor, and overall quality of our work. Below, we provide a detailed point-by-point response addressing each of your comments.
Comment 1:
I don’t see a clear explanation of how the literature was selected. There’s no mention of which databases were used or how many papers were initially found.
Response 1:
We fully agree with the reviewer that transparency regarding the search process is essential to clarify the focus and promote reproducibility. Therefore, we have now added a detailed description of our search strategy, including the selection procedure, to the methodology section (Lines 116–121).
Comment 2:
Although the focus is supposed to be on drivers, many examples come from hospital, office, or wheelchair contexts. This weakens the relevance for the transportation domain.
Response 2:
We thank the reviewer for this important remark. We have now added a dedicated paragraph to the discussion section (Lines 546-560) that explicitly addresses the transferability of results from non-driving contexts (e.g., medical or office environments) to driving scenarios. This addition clarifies the potential and the limitations of applying such findings to driver monitoring and accident prevention.
Comment 3:
Table 1 contain a large amount of valuable information, but they are dense and hard to interpret at a glance. I recommend improving their readability by applying clearer formatting, grouping similar entries, or highlighting key values. This would make the comparisons more accessible to the reader.
Response 3:
We thank the reviewer for this helpful suggestion. To improve readability, we have streamlined Table 1 to include only the most relevant information. Additionally, we have organized the studies into three groups: “electrical sensors” (top), “visual sensors” (middle), and “pressure sensors” (bottom), consistent with Figure 3 in the review paper. We have separated these groups by brief interruptions in the table to visually highlight the categories, making it easier for the reader to interpret and compare the entries at a glance.
Comment 4:
Several studies rely on self-reported fatigue levels (e.g., KSS), which are not reliable. I’d expect stronger critique of this limitation.
Response 4:
We thank the reviewer for this valuable observation. We have now explicitly addressed this limitation in the discussion section (Lines 599–607), emphasizing that reliance on self-reported fatigue measures, such as the Karolinska Sleepiness Scale (KSS), can introduce bias and reduce reliability. We also highlight the need for objective and complementary measures in future research to improve assessment accuracy.
Comment 5:
The discussion mostly summarizes findings without analyzing the reasons for limitations, low accuracy, or practical challenges in real-world use.
Response 5:
We thank the reviewer for this comment. We have revised the discussion section (Lines 596–637) to include a more detailed analysis of the reasons behind observed limitations, low accuracy, and practical challenges in real-world applications. This addition provides a deeper understanding of the factors affecting the performance of the reviewed methods and enhances the practical relevance of our work.
Comment 6:
The conclusion is too general. I’d prefer to see a more structured suggestion of next research steps or open technical problems.
Response 6:
We thank the reviewer for this suggestion. We have revised the conclusion section (Lines 717–731) to include a structured outline of remaining challenges and specific recommendations for future research. This revision highlights open technical problems and provides clear guidance for subsequent studies in the field.
Round 2
Reviewer 1 Report
Comments and Suggestions for Authors
Title: Analysis of Physiological Parameters and Driver Posture for
Prevention of Road Accidents: A Review
The authors have addressed several comments and recommendations. They have highlighted the relevance in the context and gone deeper into the research opportunities. It includes additional referred papers to support the state of the art of the topic.
The review is interesting and gives readers a glimpse into the topic. But as authors recognize, it is not a deeper analysis of the results because they are not always comparable. Therefore, this manuscript could contribute to encouraging further studies in the combined analysis of these two aspects considered in the inference of risky conditions as a preventive approach to road accidents due to fatigue or atypical health conditions.
Author Response
Thank you very much for your feedback. We will consider your ideas and take your suggestions into account for future improvements.
Reviewer 2 Report
Comments and Suggestions for Authors
The author has addressed my concerns well. I would recommend the publication of this paper. However, the author should add more descriptions to highlight the core innovation of this work. Furthermore, the driver state monitoring on driver driving ability can refer to the recent work “Authority Allocation Strategy for Shared Steering Control Considering Human-Machine Mutual Trust Level, IEEE Transactions on Intelligent Vehicles, vol. 9, no. 1, pp. 2002-2015, Jan. 2024”.
Author Response
Thank you very much for your feedback. We considered your suggestions and added them in the introduction of our paper from line 87 to 94 and line 110 to 114.
Reviewer 3 Report
Comments and Suggestions for Authors
Thanks for addressing my comments.
Author Response
We sincerely thank the reviewer for taking the time to re-evaluate our manuscript and for confirming that their comments have been addressed. We appreciate the valuable feedback provided during the review process, which has helped us to improve the quality and clarity of our work. We carefully revised the manuscript and improved the clarity and readability of the text throughout.